# Avoiding Misdiagnosis in Global Rostral Midbrain Syndrome (GRMS): Clinical Insights and Neurorehabilitation Approaches

**DOI:** 10.3390/jcm13195752

**Published:** 2024-09-27

**Authors:** Jane Jöhr, Aurea Alioth, Sabina Catalano Chiuvé, Sameer Nazeeruddin, Amani Belouaer, Roy Thomas Daniel, Shahan Momjian, Karin Diserens, Julien F. Bally

**Affiliations:** 1Service of Neurology, Department of Clinical Neurosciences, Lausanne University Hospital (CHUV) and University of Lausanne (UNIL), 1011 Lausanne, Switzerland; karin.diserens@chuv.ch (K.D.); julien.bally@chuv.ch (J.F.B.); 2Service of Neurology, Department of Clinical Neurosciences, Geneva University Hospital, 1211 Geneva, Switzerland; aurea.moren@hug.ch (A.A.); sabina.catalano@hug.ch (S.C.C.); 3Service of Neurosurgery, Department of Clinical Neurosciences, Lausanne University Hospital (CHUV) and University of Lausanne (UNIL), 1011 Lausanne, Switzerlandamani.belouaer@hcuge.ch (A.B.); roy.daniel@chuv.ch (R.T.D.); 4Service of Neurosurgery, Department of Clinical Neurosciences, Geneva University Hospitals (HUG) and University of Geneva, 1205 Geneva, Switzerland; shahan.momjian@hcuge.ch

**Keywords:** aqueductal stenosis, hydrocephalus, neurobehavioral evaluation, coma, disorder of consciousness, clinical diagnosis, case series

## Abstract

This study reports two cases of Global Rostral Midbrain Syndrome (GRMS) and corpus callosum infarction in the context of shunt overdrainage caused by obstructive hydrocephalus due to aqueductal stenosis. We detail how thorough clinical evaluation and appropriate investigation helped avoid a coma misdiagnosis and describe the excellent response to pharmacological treatment and successful neurorehabilitation in both cases. We analyze the cognitive profile of patients with GRMS, a rare condition that mimics disorders such as coma and progressive supranuclear palsy at various stages. In conscious cases, GRMS typically presents with parkinsonian syndrome, Parinaud syndrome, and cognitive issues. The awareness of this rare complication of shunt overdrainage facilitates more accurate diagnosis and better management.

## 1. Introduction

In cases of shunt treatment for obstructive hydrocephalus due to aqueductal stenosis, shunt malfunction can, in rare instances, lead to the development of Global Rostral Midbrain Syndrome (GRMS). The clinical presentation of GRMS is complex and may include multiple syndromes, as defined by Barrer et al. [1], such as parkinsonian signs accompanied by Parinaud syndrome (paralysis of upward and/or downward gaze), pyramidal signs, cognitive disorders (primarily memory-related based on the few described cases), akinetic mutism, and, in severe cases, significant impairment of wakefulness, potentially leading to coma [1,2,3,4,5,6]. This syndrome typically follows the underdrainage of the ventricles but can also occur after overdrainage, which may also damage the corpus callosum [4]. Overdrainage causes the third and lateral ventricles to collapse, pulling the midbrain upwards through the tentorial notch and the corpus callosum downwards. The parkinsonian syndrome of GRMS can be explained by damage to the nigrostriatal dopaminergic fibers, caused by the compressive effect of hydrocephalus in the upper midbrain; treatment is based on the administration of levodopa. The rare occurrence of GRMS and the potential for a fatal outcome without appropriate treatment necessitate an early identification of clinical signs through thorough neurobehavioral examination followed by a comprehensive assessment of cognitive disorders to propose suitable cognitive rehabilitation. Here, we report and analyze two cases of GRMS in the context of shunt overdrainage. The study aims to demonstrate how thorough clinical evaluation and appropriate investigation can help to avoid misdiagnosis, particularly the misdiagnosis of coma, and to highlight the importance of neurorehabilitation in successfully managing these cases. Additionally, the study seeks to contribute to the existing body of knowledge by examining the cognitive profiles of patients with GRMS and proposing effective treatment strategies.

## 2. Case Reports

### 2.1. Case 1

A 50-year-old woman with a history of a ventriculoperitoneal (VP) shunt for Sylvian aqueduct stenosis presented to the emergency department with a sudden onset of psychomotor slowing and dysarthria but without accompanying symptoms such as headache, nausea, or vomiting. Her medical history included a subarachnoid hemorrhage (Fischer I, WFNS I) from a ruptured basilar artery tip aneurysm eight years earlier, which had been treated endovascularly. This was later complicated by a cerebellar abscess requiring surgical evacuation two months after the initial treatment. Thirteen months before her current admission, she developed obstructive hydrocephalus due to Sylvian aqueduct stenosis, necessitating a ventriculocisternostomy. This procedure was complicated by the onset of iatrogenic diabetes insipidus. When the ventriculocisternostomy proved non-functional, a VP shunt with a setting of 100 mm H_2_O was implanted, leading to an uneventful clinical course until her current presentation.

A brain CT scan performed upon admission did not reveal a shunt disconnection, but a brain MRI indicated triventricular hydrocephalus (Figure 1A). A shunt revision was performed, revealing no malfunction, with intra-abdominal pressure measured at 11 cm H_2_O. Despite the increased intra-abdominal pressure due to laparoscopic gas, cerebrospinal fluid (CSF) drainage into the peritoneum was observed. However, one week later, the patient presented with psychomotor retardation, dysarthria, headache, and sleepiness. A new surgery was performed to remove the entire shunt and place an external ventricular derivation (EVD) for better CSF drainage control. This resulted in symptom resolution, the disappearance of headache, and the improvement of psychomotor retardation, leading to the internalization of the drain.

Due to low intracranial pressure monitored via the EVD, a decision was made to perform a ventriculoatrial derivation with the valve set at 60 mm H_2_O. Four days post-operatively, her consciousness deteriorated, and a brain MRI showed collapsed ventricles with signs of overdrainage (Figure 1B), prompting a change in shunt settings. New imaging revealed progressive ventricular expansion, hematic changes associated with the drain’s passage through the splenium of the corpus callosum and left posterior thalamus, a periaqueductal mesencephalic lesion, and bilateral cerebellar ischemic scars (Figure 1C). Over three weeks, the patient underwent five cycles of ventricular dilation and collapse. Endocrinologically, she presented with iatrogenic decompensation of her diabetes insipidus.

Neurobehavioral examination using standardized scales, such as the Glasgow Coma Scale (GCS) and the Coma Recovery Scale—Revised (CRS-R) [7], categorized her as being in a comatose state, with no eyelid opening or voluntary motor response to stimulation. However, a detailed clinical assessment using the complementary Motor Behavior Tool—revised (MBTr) and a systematic search for motor interaction evidence revealed clear signs of conscious perception and interaction, such as grimacing in response to pain and intentional defensive responses to painful stimuli [8,9].

The discrepancy between the CRS-R’s conclusion of a disorder of consciousness and the motor behavior observations, combined with the clinical picture of parkinsonian signs, Parinaud syndrome, and akinetic mutism, led to a suspicion of Global Rostral Midbrain Syndrome (GRMS). A test with subcutaneous apomorphine infusion was proposed. The patient was transferred to the Acute Neurorehabilitation Unit for six weeks before being referred to a post-acute neurorehabilitation center.

Physical examination upon entering the unit revealed bilateral relative mydriasis, gaze paresis with chronic sunset eyes without gaze tracking, bilateral upper quadrantanopia, a rigid hypertonia of the wrists with paratonia on the right, a hypertonia of the ankles with equinus feet and incomplete dorsiflexion, and symmetrical increased osteotendinous reflexes extending to all four limbs. The effective apomorphine infusion improved alertness and allowed a transition to oral dopaminergic treatment (Levodopa and Benserazide, 200/50 mg 4×/day). This further improved her clinical condition, eliminating parkinsonian signs and progressively normalizing vigilance, enabling coherent conversation.

Neuropsychological evaluation three weeks after entry revealed severe bimodal anterograde memory impairment (episodic, encoding, and recovery strategies) and severe processing speed and executive dysfunction (cognitive/behavioral/emotional). Neuropsychological priorities included strengthening temporal cues, setting up a memory diary, familiarizing the patient with mnemonic strategies, increasing reactivity and attentional endurance, and enhancing language productivity and conversational dynamics.

Remote imaging one month later showed signs of Wallerian degeneration extending to the splenium of the corpus callosum, the stable scalloped appearance of the corpus callosum, and the focal atrophy of the anterior corpus callosum and isthmus (Figure 2). PET imaging revealed mild cortical hypometabolism in the bilateral prefrontal and inferior parietal areas, sparing the sensorimotor cortex, and asymmetric cerebellar uptake favoring the left side. Three months after the transfer to the rehabilitation center, the patient returned home. A follow-up visit nine months after hospitalization showed favorable motor and functional progress. The oral dopaminergic treatment had been discontinued without worsening motor parkinsonian symptoms; the lack of motivation, or apathy, was likely due to lesions in the dopaminergic pathways from the ventral tegmental area to the limbic striatum and medial frontal cortex.

### 2.2. Case 2

A 63-year-old woman underwent a ventriculoperitoneal (VP) shunt procedure for triventricular hydrocephalus due to Sylvian aqueduct stenosis (Figure 3A,B). Four months after VP shunt implantation, she developed psychomotor retardation and cognitive disorders, particularly memory impairment, along with apathy and difficulty walking. A few months later, she developed vertical gaze palsy. CT and MRI scans (Figure 3C,D) showed a shrinkage of the ventricular system, resulting in slit ventricles, an elongation of the midbrain, and callosal infarction. Although valve settings were adjusted, this rapidly led to the reformation of slit ventricles. After three cycles of dilated and collapsed ventricles over ten days and difficulty in finding the correct drainage setting, the patient received an assist shunt, which finally normalized the ventricle size and restored normal vigilance, though vertical gaze palsy persisted.

However, just four days after the assist shunt implantation, she presented again in a comatose state due to ventricular system dilation. Over the next two months, she underwent multiple valve setting adjustments, leading to cycles of dilated and collapsed ventricles. Eventually, her vigilance stabilized, but her cognitive functions and walking capabilities progressively deteriorated. Neurologically, she gradually exhibited parkinsonism with tremor, bradykinesia, axial rigidity, eyelid-opening apraxia, and persistent supranuclear gaze palsy. The combination of vertical gaze paralysis and parkinsonian symptoms resembled progressive supranuclear palsy (see Figure 4 for a comprehensive timeline).

A decision was made to replace the entire ventricular system with a fixed, constant-flow valve, which finally stabilized the ventricular system. In the context of severe parkinsonism, likely due to lesions in the nigrostriatal pathways confirmed by a positive (123)I-FP-CIT SPECT (DaTSCAN^®^), symptomatic dopaminergic treatment (Levodopa and Benserazide, 100/25 mg 4×/day) was initiated alongside intensive neuro-rehabilitation in a post-acute rehabilitation center for 22 weeks.

Upon entry, the physical examination revealed normal pupil reactivity, smooth and complete horizontal gaze pursuit, but impaired vertical gaze pursuit, symmetrical sensitivity in all four limbs, parkinsonian rest tremor in the right hand, and rigid hypertonia, slightly predominant on the left. Her feet were in total plantar extension, non-reducible on the left and slightly (about 20°) on the right. There was progressive clinical improvement, primarily in mobility in all four limbs, with a consequent reduction in axial and appendicular rigidity. While the dopaminergic treatment likely contributed significantly to clinical improvement and the stabilization of intracranial pressure, and hydrocephalus also played a crucial role.

Neuropsychological assessment at admission indicated severe cognitive impairment in all areas, with complete dependence for activities of daily living. A follow-up CT scan at two months showed increased signs of ventricular overdrainage, with a collapsed appearance of the lateral and third ventricles but no complications. A neurosurgical opinion was sought; however, in the reassuring clinical context, with the patient showing no new vigilance disorders, no intervention was proposed. An injection of botulinum toxin into the tricep muscles (right and left) resulted in slight clinical improvement.

Approximately four months after her transfer to the rehabilitation center, the patient returned home with at-home physiotherapy sessions. At discharge, neuropsychological assessment revealed moderate to severe executive dysfunction, significant verbal episodic memory impairment, moderate non-lateralized attentional disorders, mild language difficulties, and impaired performance on social cognition tasks. A follow-up assessment five months later showed an improved neuropsychological profile, though severe episodic memory impairment and moderate executive difficulties remained. Notably, stopping the dopaminergic treatment after the neuro-rehabilitation period did not cause the parkinsonian syndrome to reoccur, despite the DaTSCAN showing a marked decrease in presynaptic dopamine transporter binding.

## 3. Discussion

We report two cases of patients who experienced shunt dysfunction with a loss of ventricular compliance, resulting in alternating between slit ventricles (overdrainage) and hydrocephalus (underdrainage) within a short period. This led to a clinical presentation characterized by parkinsonian syndrome, akinetic mutism, Parinaud syndrome, cognitive disturbances, pyramidal signs, and symptoms that standard tools might have interpreted as coma—all signs characteristic of Global Rostral Midbrain Syndrome. We discuss their clinical presentation, highlighting the danger of underestimating their actual level of consciousness and describing the cognitive impairment in the setting of hydrocephalus and corpus callosum infarction and appropriate neurorehabilitation management.

In our two patients, the profiles of cognitive impairment were not identical, though certain features overlapped. These included deficits affecting memory (impaired learning and retrieval of information), executive functions (task initiation and pursuit, conceptualization, and mental flexibility), and attention (support and processing speed). This pattern of memory impairment appears to be associated with obstructive hydrocephalus following aqueductal stenosis, resulting in ventricular distress with repercussions on adjacent structures.

According to Donnet and colleagues [10], memory impairment in this type of hydrocephalus correlates with the enlargement of the third ventricle, whose functional environment is involved in memory processes, motor and executive control, and endocrine and vegetative regulation. They report a significant correlation between the width of the third ventricle and deficits in immediate and delayed recall of a story, learning and delayed recall of a word list, and verbal fluency in patients with hydrocephalus due to aqueductal stenosis. These deficits should be interpreted in the context of damage to the fornix and its connections with the hippocampus and mammillary bodies, structures involved in memory processes.

In case 1, the patient’s verbal episodic memory performance reveals a deficit mainly in encoding and the implementation of retrieval strategies but optimal delayed term-to-term recognition, suggesting information retention. These results are consistent with those described in the literature, showing a relative preservation of recognition abilities compared to recall abilities in patients with limbic system lesions [11].

Regarding executive function, we found severe impairment in our two patients, affecting the main functions of inhibition, flexibility, and working memory updating, supporting higher-level mechanisms such as reasoning and planning. Our results align with those described in the literature in patients with hydrocephalus, who typically show significant executive function impairment [12]. Donnet et al. found that executive functions in patients with normal-pressure hydrocephalus were more severely impaired than in those with aqueductal stenosis [10]. The frontal lobe, a primary brain structure related to executive functions, would be involved in hydrocephalus due to subsequent subcortical lesions damaging the fronto-subcortical loops [13].

The attentional difficulties observed in our patients—severe psychomotor slowing, marked intellectual fatigability, difficulties in maintaining sustained cognitive effort, self-distractibility, and overt signs of exertion on all tests—are frequently described in hydrocephalus patients [12]. In the context of GRMS developed by our two patients, another explanation for the cognitive and behavioral disorders could be damage to dopaminergic pathways located in the ventral midbrain, including the mesolimbic, mesocortical, and nigrostriatal pathways (noted by the positive DaTSCAN in case 2). These pathways regulate several physiological functions such as motivational behavior, behavioral fragment selection (both motor and non-motor), emotional response, and learning and memory [14]. Following the multiple expansion and collapse of the third ventricle, lesions of these pathways can lead to dopaminergic depletion, resulting in rigidity and a massive reduction in speech and voluntary behavior (akinetic mutism).

The cognitive and behavioral symptoms in patients with GRMS are similar to those observed in patients with cerebellar cognitive affective syndrome (CCAS) [15], a condition resulting from diaschisis linked to disruption of the circuits uniting the cerebellum with the prefrontal, posterior parietal, superior temporal, and limbic cortices [16]. This disruption causes a constellation of deficits in executive function, spatial cognition, language, and affectivity. Schmahmann describes disturbances in visuospatial organization, processing speed, planning and programming abilities, verbal fluency, resistance to interference, and working memory as prominent in these patients [17]. Our patient’s case, with lesions in both regions, suggests a common involvement of these structures in the expression of her disorders.

These are two particularly rare cases of complications arising from drain malfunction in the context of obstructive hydrocephalus. However, their rarity should not undermine the importance of a thorough neurobehavioral assessment in the acute phase when consciousness disturbances are present (or when the absence of overt interaction suggests so). Indeed, the initial clinical assessment of these two patients, using standardized scales such as the GCS or CRS-R, led to a diagnosis of coma, resulting in an underestimation of the state of consciousness and an unfavorable prognosis. However, an in-depth neurobehavioral assessment looking for subtle signs of conscious perception, as well as pitfalls limiting motor interaction, enabled us to characterize the presence of akinetic mutism accompanied by vigilance disorders, resulting in a clinical situation mimicking merely a disorder of consciousness.

Following the model described by other authors [4], we administered dopaminergic treatment (apomorphine infusion and/or levodopa), to which the two cases presented here demonstrated an excellent response. This aligns with findings from Zhang et al., where a case of parkinsonism following ventriculoperitoneal shunt showed marked improvement with high-dose levodopa. Their study underscores the role of dopaminergic pathways in the recovery of motor function in similar cases of hydrocephalus-induced parkinsonism. This supports the hypothesis that early intervention with dopaminergic drugs can lead to substantial clinical improvements in patients with GRMS [6].

Neurorehabilitation focused on the cognitive and functional aspects of the deficit, starting as early as possible, is recommended. However, the cognitive management of GRMS syndrome is not clearly defined in the literature. Zaksaite et al. highlight the lack of studies devoted to the efficacy of hydrocephalus-specific cognitive rehabilitation interventions [12]. In general, practice recommendations for managing cognitive impairment following hydrocephalus align with those for TBI and stroke [18]. Through interventions targeting global cognitive processes, generalization to higher-order cognitive functions could be established. Cognitive rehabilitation for patients with hydrocephalus targets processing speed and attentional endurance, metacognitive training, and the development of compensatory strategies. For memory, it includes error-free learning, memorization strategies, and the use of external aids [19].

GRMS is a rare complication primarily associated with shunt malfunction in patients with hydrocephalus. The syndrome has been described in a limited number of case reports, with the majority of these highlighting its occurrence following shunt underdrainage. However, instances of GRMS following shunt overdrainage, as reported in our cases, are even more uncommon. Barrer et al. reported one of the earliest cases of GRMS secondary to shunt malfunction [1]. The authors described a clinical presentation of parkinsonian symptoms, Parinaud’s syndrome, and coma-like states in patients with shunt underdrainage. The focus was primarily on the recognition of GRMS as a differential diagnosis in shunt-related complications. Cinalli et al. expanded on the understanding of GRMS by analyzing multiple cases of Sylvian Aqueduct Syndrome [2]. This study emphasized the correlation between shunt malfunction and midbrain dysfunction, particularly noting the importance of early shunt revision to prevent permanent neurological damage. Villamil et al. highlighted GRMS cases associated with shunt overdrainage, similar to our cases [4]. Their report underlined the challenge of diagnosing GRMS in the acute setting due to its clinical overlap with other disorders of consciousness. They also advocated for immediate corrective surgery and the initiation of dopaminergic treatment as key steps in management. Zhang et al. described a case of parkinsonism following ventriculoperitoneal shunt, where symptoms worsened with antipsychotic medication but responded well to Madopar. This case emphasized the importance of recognizing reversible dysfunction of the presynaptic nigrostriatal dopaminergic pathway, consistent with our treatment approach [5]. Additionally, Zhang et al. reported another case of levodopa-responsive parkinsonism with akinetic mutism following ventriculoperitoneal shunt, highlighting the role of dopaminergic therapy in reversing brain hypometabolism and improving clinical outcomes [6]. These cases further support the effectiveness of dopaminergic therapy in GRMS management, as demonstrated in our cases.

Our study adds to the limited body of evidence by presenting two cases of GRMS following shunt overdrainage. Unlike previous reports, our detailed neurobehavioral assessment enabled the early detection of consciousness in patients who otherwise appeared comatose. This underscores the importance of comprehensive clinical evaluation and the use of advanced assessment tools like the MBTr. Furthermore, the successful neurorehabilitation outcomes in our cases highlight the potential for recovery with appropriate and timely interventions. Our findings are novel in that they demonstrate the effectiveness of early intervention with dopaminergic treatment and intensive neurorehabilitation in reversing severe symptoms of GRMS, including akinetic mutism and parkinsonian syndrome. The successful recovery of both patients, as documented in our study, suggests that with timely and appropriate management, even severe cases of GRMS can lead to significant functional improvement. This highlights the critical need for awareness and early detection of this syndrome among clinicians, which could improve patient outcomes in similar future cases. Furthermore, this study contributes to the understanding of cognitive impairments associated with GRMS, particularly in relation to the effects of hydrocephalus on memory, executive function, and attention. By drawing connections between the clinical presentation of GRMS and cerebellar cognitive affective syndrome, our report opens avenues for future research on the overlapping cognitive and behavioral symptoms in these conditions. The management overview for the two cases is summarized in Table 1.

### 3.1. Limitations

One limitation of this report is the small number of cases, which adds to the rare instances described in the literature; thus, the recommendations are not necessarily generalizable to all patients with similar characteristics. Another limitation is that these two cases come from different hospitals (although the movement disorders specialist was the same), with differing cognitive assessment and management practices in the acute phase, possibly impacting the cognitive and functional evolution of the patients.

### 3.2. Future Directions

Future research should aim to expand the understanding of GRMS, particularly regarding neurorehabilitation strategies that can optimize recovery. Ongoing clinical trials are essential to evaluate the efficacy of various pharmacological interventions, such as dopaminergic treatments, in managing the cognitive and motor impairments associated with GRMS. Additionally, studies investigating the long-term outcomes of GRMS patients and their quality of life following different therapeutic approaches could offer valuable insights into the most effective treatment modalities. Collaborative research efforts across multiple centers could help to establish standardized guidelines for the treatment and rehabilitation of GRMS, ensuring that patients receive the most effective care possible. These initiatives would contribute to a more comprehensive evidence base, ultimately leading to improved clinical outcomes for patients affected by this rare but severe syndrome.

## 4. Conclusions

The two cases reported here underscore the critical importance of an early identification of clinical signs of conscious perception in patients with GRMS through detailed neurobehavioral assessment. Our findings show that early pharmacological intervention with dopaminergic treatments, combined with timely neurorehabilitation, can result in significant improvements in motor and cognitive functions. This study highlights the role of comprehensive clinical evaluation in avoiding misdiagnosis, particularly in cases that mimic coma, and emphasizes that prompt therapeutic action is essential to prevent permanent neurological damage. These results advocate for the integration of advanced assessment tools and early neurorehabilitation protocols in clinical practice, improving the outcomes for patients affected by this rare but severe condition.

## Figures and Tables

**Figure 1 jcm-13-05752-f001:**
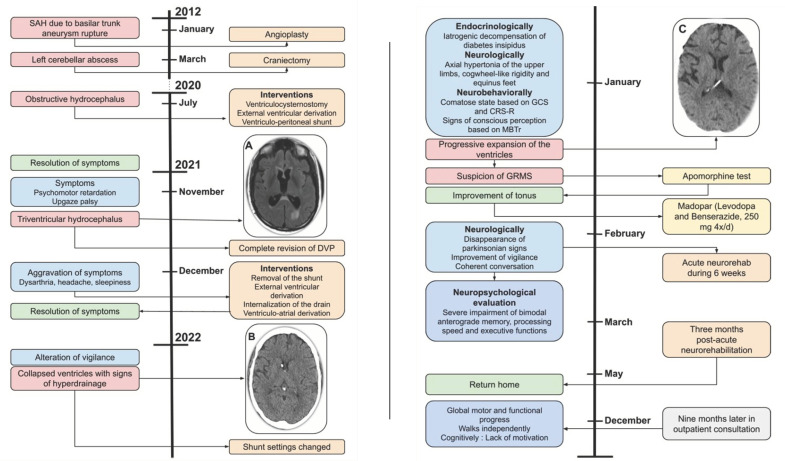
Case 1 timeline.

**Figure 2 jcm-13-05752-f002:**
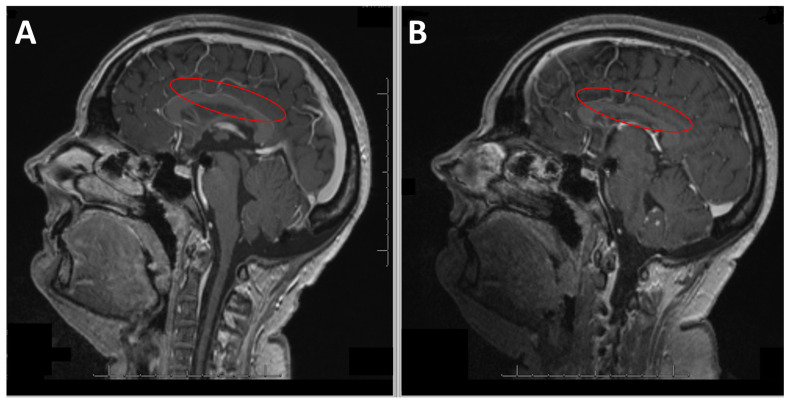
Case 1: (**A**) Sagittal T1-weighted MRI showing the undulating appearance of the corpus callosum (red oval on (**A**)) described as scalloping indicating a segmental ventral collapse between the branches of the pericallosal artery connecting the corpus callosum to the cingulate gyrus secondary to hydrocephalus after VP shunting. (**B**) Sagittal T1-weighted MRI showing atrophy of the corpus callosum (red oval on (**B**)).

**Figure 3 jcm-13-05752-f003:**
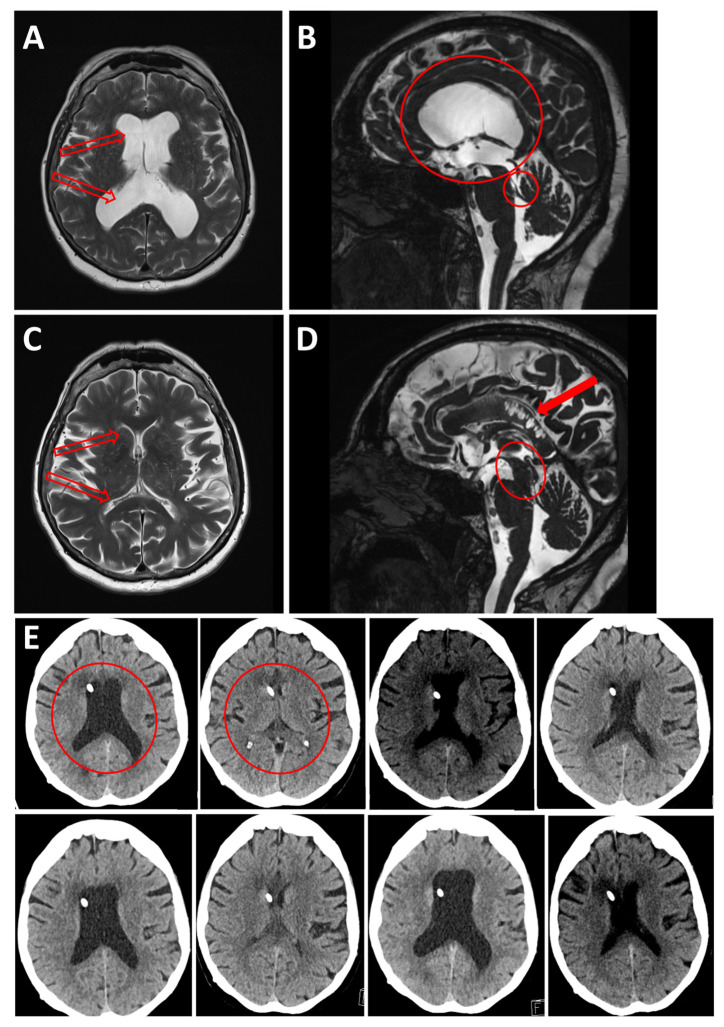
Case 2: (**A**) Axial and (**B**) Sagittal T2-weighted MRI showing triventricular hydrocephalus (red empty arrows on (**A**); large red circle on (**B**)) due to Sylvian aqueduct stenosis (small red circle on (**B**)). (**C**) Axial and (**D**) Sagittal T2-weighted MRI showing shrinkage of the ventricular system, resulting in slit ventricles (red empty arrows on (**C**)), elongation of the midbrain (red oval on (**D**)) and callosal infarction (full red arrow on (**D**)) due to its compression against the falx. (**E**) Illustration of the rapid alternating ballooning (e.g.: red circle on first **upper left panel**) and shrinking (e.g.: red circle on second **upper left panel**) of the ventricles over a short period of time (September through November 2019), due to alternating overdrainage and underdrainage. First image, upper left: four days post shunt placement. The following images represent the different adjustments of valve pressure.

**Figure 4 jcm-13-05752-f004:**
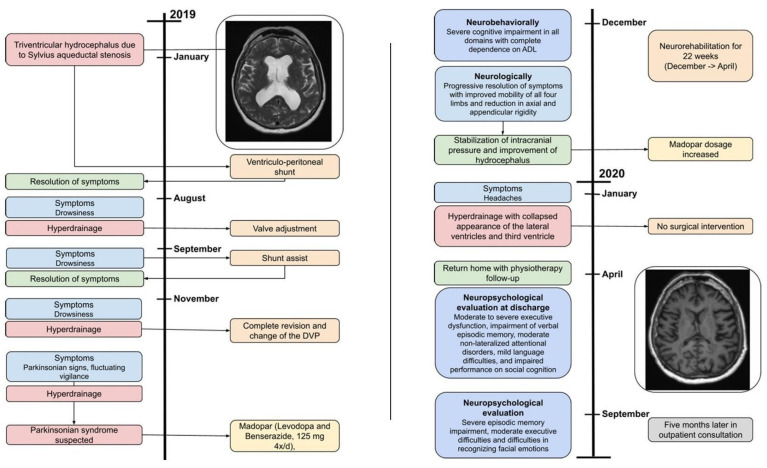
Case-2 Timeline.

**Table 1 jcm-13-05752-t001:** Management overview for the two cases.

Management Aspect	Details
**Treatment of Shunt Dysfunction**	**Initial Diagnosis and Intervention**: Imaging techniques (CT, MRI) were used to diagnose shunt dysfunction, leading to adjustments in shunt settings. **Surgical Interventions**: External ventricular derivation and ventriculo-peritoneal and ventriculo-atrial shunt placement were performed to stabilize intracranial pressure. **Outcome**: Shunt revision managed hydrocephalus, requiring multiple adjustments for optimal results.
**Proposed Treatment Strategy for GRMS**	**Neurobehavioral Assessment**: Utilized CRS-R and MBTr tools for early detection of consciousness, avoiding misdiagnosis of coma. **Pharmacological Management**: Apomorphine infusion and oral levodopa were used to treat parkinsonian symptoms. **Recommendation**: Early assessment, pharmacological intervention, and intensive neurorehabilitation are crucial.
**Management of Parkinsonism**	**Pharmacological Treatment**: Dopaminergic medications (Levodopa, Benserazide) effectively managed motor symptoms and improved cognitive alertness. **Neurorehabilitation**: Focused on enhancing motor function, cognitive performance, and daily living activities. **Outcome**: Significant improvement in motor and cognitive functions with sustained treatment and rehabilitation.
**Recommendations to Avoid GRMS**	**Shunt Management**: Regular monitoring of shunt function and intracranial pressure to prevent complications. **Early Intervention**: Immediate adjustment of shunt settings and initiation of dopaminergic therapy upon detecting GRMS signs. **Long-Term Follow-Up**: Regular neurobehavioral assessments and follow-up for early detection and management of symptoms.

## Data Availability

The data in this study are available on request from the corresponding author. The data are not publicly available due to privacy restrictions.

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
