# Peer review of "Avoiding Misdiagnosis in Global Rostral Midbrain Syndrome (GRMS): Clinical Insights and Neurorehabilitation Approaches"

_jcm, 2024, doi:10.3390/jcm13195752_

Round 1

Reviewer 1 Report

Comments and Suggestions for Authors

This paper reports 2 particularly well described cases of GRMS. Although it does not contain strictly speaking new elements.  On the one hand this rare syndrome must be the subject of such detailed descriptions for it to be well recognized by practitioners, on the other hand it includes clinical analyzes which make it possible to establish the differential diagnosis, of the highest importance, with a coma in particular.

The paper is very well written, the iconography and timelines excellent, and the discussion is well structured. I don't have any suggestions for significant changes to make.

Author Response

Comment : The paper is very well written, the iconography and timelines excellent, and the discussion is well structured. I don't have any suggestions for significant changes to make.

Response : We warmly thank the reviewer for reading our manuscript and for their appreciation of our work.

Reviewer 2 Report

Comments and Suggestions for Authors

The authors presented two cases of GRMS, of impact by its rare occurrence. Nevertheless, some issued should be addressed:

1) The objectives of the study should be presented in the introduction section;

2) The ethical aspects of the study mentioned briefly (mainly by Helsinki declaration) and should be expanded (e.g., the informed consent)

3) The authors should mention the impact of the findings on the existing body of evidence, highlighting the novelty of the manuscript;

4) Some future directions or ongoing clinical trials to improve patients outcomes should be mentioned.

Comments on the Quality of English Language

Minor English editing required

Author Response

Comments 1 : The objectives of the study should be presented in the introduction section

Response 1 : Thank you for your valuable feedback. We agree with your suggestion and have revised the introduction section accordingly to clearly present the objectives of the study. The changes were made on page 2, paragraph 1 and line 46. 

Comments 2 : The ethical aspects of the study mentioned briefly (mainly by Helsinki declaration) and should be expanded (e.g., the informed consent)

Response 2 : Thank you for your suggestion. We have expanded the ethical aspects section to include more details on dates of approval for informed consent and other relevant considerations. The changes were made on page 12, and line 384.

Comments 3 : The authors should mention the impact of the findings on the existing body of evidence, highlighting the novelty of the manuscript;

Response 3 : Thank you for your valuable comment. We have revised the manuscript to clearly address the impact of our findings on the existing body of evidence and highlight the novel contributions of our study. The changes were made on page 11 and line 337.

Comments 4 : Some future directions or ongoing clinical trials to improve patients outcomes should be mentioned.

Response 4 : Thank you for your suggestion. We have included a section discussing potential future directions and ongoing clinical trials that aim to improve patient outcomes. The changes were made on page 11 and line 357.

Comments 5 on the Quality of English Language : Minor changes. 

Response 5 : Thank you for your feedback regarding the quality of the English language. We have carefully reviewed the manuscript and made the necessary minor revisions to improve clarity and readability. We believe the changes have enhanced the overall quality of the text.

Reviewer 3 Report

Comments and Suggestions for Authors

Authors present  two cases of on global rostral midbrain syndrome (GRMS) which may mimic coma, and reflect on avoiding misdiagnosis and promoting neurorehabilitation. Authors reflect and describe in detail two cases of shunt overdrainage, which lead to upward herniation of midbrain and coma with parkinsonism. This is an important differential diagnosis when it comes to VP shunt insufficiency or overdrainage in cases of obstructive hydrocephalus.

We recommend that authors add several more figures with CTs in both cases to follow the outline of the events. Also, management of the patients should be made in point-to-point overview: what was the treatment of shunt dysfunction and how was it treated, what do authors propose as a treatment in these cases, was was the treatment of parkinsonsism, what do authors recommend to avoid GMRS. 

Include a literature review with analysis of the few cases described. 

Comments on the Quality of English Language

Minor changes. 

Author Response

Comments 1 : We recommend that authors add several more figures with CTs in both cases to follow the outline of the events.

Response 1 : Thank you for your suggestion. We agree that additional CT images would enhance the clarity of the clinical timeline and the understanding of the phenomenon. We have included several more figures with CT scans to better illustrate the progression of events and the corresponding changes observed in case 2. Changes were made on page 7.

Comments 2 : Also, management of the patients should be made in point-to-point overview: what was the treatment of shunt dysfunction and how was it treated, what do authors propose as a treatment in these cases, was was the treatment of parkinsonsism, what do authors recommend to avoid GMRS. 

Response 2 : Thank you for your detailed feedback. We agree that a point-by-point overview of the patient management would provide clearer guidance for clinicians. We have revised the manuscript to include a table with a structured overview that outlines the treatment steps for shunt dysfunction, our proposed approach for managing similar cases, the treatment of parkinsonism, and recommendations to avoid the development of GRMS. This will help to ensure that the key management strategies are clearly communicated and easily accessible to the readers. The table has been added on page 12, and line 370.

Comments 3 : Include a literature review with analysis of the few cases described. 

Response 3 : Thank you for your suggestion. We agree that including a literature review with an analysis of the few cases described would enhance the context and depth of our study. We have added a dedicated section in the discussion that summarizes and compares the existing literature on GRMS, particularly focusing on cases associated with shunt malfunction. This section highlights the similarities and differences between the reported cases and our findings, further emphasizing the novelty and significance of our study. The changes were made on page 10 and line 316.

Comments 4 on the Quality of English Language : Minor changes. 

Response 4 : Thank you for your feedback regarding the quality of the English language. We have carefully reviewed the manuscript and made the necessary minor revisions to improve clarity and readability. We believe the changes have enhanced the overall quality of the text.

Round 2

Reviewer 3 Report

Comments and Suggestions for Authors

Authors have sufficiently responded. 

Author Response

Comments and Suggestions for Authors : Authors have sufficiently responded. 

Response : Thank you for taking the time to review our work and for your valuable feedback throughout the process. We are glad to hear that our responses sufficiently addressed your comments and suggestions.